# Detecting Early-Stage Oral Cancer from Clinically Diagnosed Oral Potentially Malignant Disorders by DNA Methylation Profile

**DOI:** 10.3390/cancers14112646

**Published:** 2022-05-26

**Authors:** Kazuki Mori, Tomofumi Hamada, Mahiro Beppu, Hiroki Tsuchihashi, Yuichi Goto, Kenichi Kume, Hiroshi Hijioka, Keitaro Nishi, Yumiko Mishima, Tsuyoshi Sugiura

**Affiliations:** 1Department of Maxillofacial Diagnostic and Surgical Science, Field of Oral and Maxillofacial Rehabilitation, Graduate School of Medical and Dental Sciences, Kagoshima University, Kagoshima 890-8544, Japan; k4639792@kadai.jp (K.M.); mbeppu@dent.kagoshima-u.ac.jp (M.B.); tsuchiha@hotmail.co.jp (H.T.); ygoto@dent.kagoshima-u.ac.jp (Y.G.); kkume@dent.kagoshima-u.ac.jp (K.K.); zio@dent.kagoshima-u.ac.jp (H.H.); knishi@dent.kagoshima-u.ac.jp (K.N.); k5782265@kadai.jp (Y.M.); 2Department of Oral & Maxillofacial Surgery, Hakuaikai Social Medical Corporation, Sagara Hospital, Kagoshima 892-0833, Japan

**Keywords:** oral cancer, oral squamous carcinoma, oral potentially malignant disorders, gargle fluid, DNA methylation, biomarker

## Abstract

**Simple Summary:**

Clinically, early-stage oral cancers are difficult to distinguish from oral potentially malignant disorders (OPMDs) because they show a variety of mucosal pathologies. Therefore, invasive tissue biopsies should be performed to determine the treatment strategy. Previously, we focused on gargle fluid as a noninvasive testing method and reported aberrant methylation in gargle fluid in patients with oral cancer. In this study, we successfully identified aberrantly methylated genes in early-stage oral cancer and reported that a combination of methylation of six genes could distinguish early-stage oral cancer from OPMDs, with high diagnostic performance. In addition, the methylation panel more accurately reflected the presence of early-stage oral cancer than cytology testing. Our results suggest that the methylation panel using gargle fluid has the potential to be used as a noninvasive screening tool to diagnose early-stage cancer.

**Abstract:**

Clinically, early-stage oral cancers are difficult to distinguish from oral potentially malignant disorders (OPMDs), and invasive tissue biopsy should be performed to determine a treatment strategy. Previously, we focused on gargle fluid as a noninvasive testing method and reported aberrant methylation in gargle fluid in patients with oral cancer. This study aimed to distinguish early-stage oral cancer from clinically diagnosed OPMDs using gargle fluid samples. We collected gargle fluid samples from 40 patients who were clinically diagnosed with OPMDs in the training set; among them, 9 patients were pathologically diagnosed with oral cancer. Methylation levels of 25 tumor suppressor genes were analyzed using the methylation-specific multiplex ligation-dependent probe amplification (MS-MLPA) method. We found that a combination of six genes (*TP73*, *CASP8*, *RARB*, *KLLN*, *GSTP1*, and *CHFR*) could distinguish oral cancer from clinically diagnosed OPMDs with high diagnostic performance (area under the curve [AUC], 0.885; sensitivity, 77.8%; and specificity, 87.1%). Additionally, the panel comprised of the six methylated genes was validated in the test set. Furthermore, when compared with cytology testing, the panel could accurately detect oral cancer. The present methylated gene panel may serve as a novel biomarker for oral cancer.

## 1. Introduction

Oral squamous cell carcinoma (OSCC) is the most common cancer occurring in the oral cavity, and according to a WHO survey, the number of new cases of lip and oral cancer worldwide in 2020 was reported to be 377,113, with 177,757 deaths. Lip and oral cancers are rare, accounting for approximately 2% of all cancers; however, mortality rates are on the rise [1]. Early detection and treatment of OSCC is important because the overall survival rate of OSCC decreases as the stage progresses, exceeding 80% in stages I and II, but approximately 60% in stages III and IV [2]. Oral potentially malignant disorders (OPMDs) have been defined as any oral mucosal abnormality that is associated with a statistically increased risk of developing oral cancer [3]. OPMDs include precancerous lesions and conditions that have been classified in a complicated manner for many years, such as leukoplakia, erythroplakia, and lichen planus. The goal of the OPMDs concept is to reach a more accurate diagnosis from the perspective of malignant or non-malignant transformation [3]. OPMDs exhibit various clinical conditions such as hypertrophic and atrophic changes; however, histopathologically, they may include lesions of epithelial hyperplasia, epithelial dysplasia, intraepithelial carcinoma, and squamous cell carcinoma. Therefore, invasive tissue biopsies are required to determine appropriate treatment strategies [3,4,5]. Other testing methods include cytology tests for the initial screening and follow-up [6,7]. Biopsies are invasive and cannot be performed repeatedly during follow-up. In contrast, cytology testing is minimally invasive, performed at many medical centers, and does not require special equipment; however, it is less sensitive [7]. Therefore, it is necessary to use other monitoring tools to objectively assess the malignant potential of OPMDs. DNA methylation, which occurs mainly at cytosines in CpG islands, regulates gene expression. In addition to genomic aberrations, epigenomic aberrations are known to play a role in the development and progression of cancer, and the relationship between cancer-related genes and aberrant methylation has been widely studied [8,9]. It has been reported that the promoter regions of cancer-related genes show abnormal hypermethylation and gene expression is often repressed, and the usefulness of DNA methylation-based diagnostic methods is being investigated in various cancers [8,10,11,12,13,14,15].

Carcinogenesis is caused by stepwise abnormalities in multiple genes, and the multistage carcinogenesis theory has been validated [16,17]. In the colorectal cancer model, many studies have been conducted based on genetic analysis, and genetic abnormalities such as *APC*, *KRAS*, and *p53* mutations are observed at each stage of adenoma, early-stage cancer, and advanced cancer [18]. The multistage carcinogenesis theory is also supported in oral cancer; however, specific genetic changes have not been elucidated. Epigenetic changes are commonly observed in OSCC, particularly aberrant methylation of tumor suppressor genes, which are implicated in carcinogenesis [19,20]. Previously, saliva, gargle fluid, and oral brushing have been examined as tools for the early detection of oral cancer using methylation as an indicator [21,22,23,24,25,26]. In our laboratory, we have focused on gargle fluid as a noninvasive testing method and have reported aberrant methylation of tumor suppressor genes in the gargle fluid of OSCC [27]. Compared with the healthy group, the OSCC group showed aberrant methylation of tumor suppressor genes in the gargle fluid, suggesting that the gargle fluid of patients with cancer contains exfoliated tumor cells and may be a biomarker for predicting the presence of cancer. DNA methylation tests have also been suggested as important sources of information to complement clinical diagnosis. In this study, we investigated the aberrant methylation of tumor suppressor genes in the gargle fluid to identify stepwise genetic abnormalities in the process of carcinogenesis and to identify patients with early-stage cancer based on this examination. In this study, we collected gargle fluid from patients with clinically diagnosed OPMDs before biopsy and analyzed the methylation of tumor suppressor genes to determine their usefulness as biomarkers for the early detection of malignant tumors, including intraepithelial carcinoma. 

Here, we used the methylation-specific multiplex ligation-dependent probe amplification (MS-MLPA) method [28,29], which can examine the methylation of 25 tumor suppressor genes that are commonly aberrantly methylated in cancers. MS-MLPA can examine the methylation of multiple genes simultaneously using a probe mix and has been used for the diagnosis of imprinted diseases such as Prader–Willi syndrome and Angelman syndrome [28]. MS-MLPA methods have been used to analyze methylation in prostate and endometrial cancers [30,31] and have also been previously used in OSCC to examine hypermethylation in tissue samples [32]. This method can potentially be used to investigate aberrant methylation in cancer, and we hypothesized that studying the methylation of tumor suppressor genes in gargle fluid is among the best methods. In anticipation of the practical application of a noninvasive test using gargle fluid as a screening tool in the future, we used the MS-MLPA method.

## 2. Materials and Methods

### 2.1. Sample Collection

We collected gargle fluid samples from 40 patients in the training set and 16 patients in the test set clinically diagnosed with OPMD at the Department of Oral Surgery, Kagoshima University Hospital (Kagoshima, Japan). The training set included patients from April 2020 to March 2021, and the test set included patients from April to October 2021. All samples were collected before tissue biopsy, and the patients were made to gargle using 20 mL of purified water for 1 min. This study was approved by the Ethics Committee on Life Sciences and Genetic Analysis, Kagoshima University Graduate School of Medical and Dental Sciences. 

### 2.2. Purification of DNA

The gargle fluid samples (20 mL) were centrifuged for 5 min at 450× *g* and 25 °C, and cell pellets were separated from the supernatant. The cell pellet was diluted with phosphate buffer saline to a total volume of 200 μL. Genomic DNA was purified from a 200-μL diluted solution using DNeasy Blood & Tissue Kit (Qiagen, Chatsworth, CA, USA) according to the protocol.

### 2.3. Methylation Analysis

The MS-MLPA method (MRC-Holland, Amsterdam, The Netherlands) was used to perform a methylation analysis for each gargle sample. ME001-D1 Tumor suppressor mix 1 (MRC-Holland, Amsterdam, The Netherlands) was used to analyze the methylation of the promoter regions of 25 tumor suppressor genes. Each MS-MLPA probe had a cleavage site for a methylation-sensitive restriction enzyme, such as Hha I. If the region targeted by the MS-MLPA probe was methylated, the restriction enzyme did not function. The targeted region remained undigested and was amplified by PCR. On the other hand, if the region targeted by the probe was not methylated, the restriction enzyme functioned. The target region was digested and unamplified by PCR. The relative peak value of each probe in the digested sample was compared with that of the undigested sample, and the peak value of the probe in the methylated region was calculated as the methylation percentage.

The probe mix was added to the DNA sample and hybridization was performed at 95 °C for 16 h. Untreated and Hha I-treated samples were prepared, and ligation and digestion were performed at 48 °C for 30 min. PCR was performed under the following conditions: (35 [95 °C for 30 s, 60 °C for 30 s, 72 °C for 60 s], 72 °C for 20 min, and 15 °C pause). The amplified fragments were analyzed using the ABI PRISM 3130XL Genetic Analyzer (ABI, MA, USA), and the peak values were calculated using Gene Mapper (ABI, MA, USA). The fragment analysis data were normalized, and the methylation percentage of each tumor suppressor gene was calculated using the analysis software Coffalyser v.140721.1958 (MRC-Holland, Amsterdam, The Netherlands).

### 2.4. Statistical Analysis

Detection of malignancy (intraepithelial carcinoma and early invasive carcinoma) was used as an endpoint, and the area under the curve (AUC) was calculated from the receiver operating characteristic (ROC) curve. Based on the sensitivity and specificity obtained from the ROC curve, a useful cut-off value for diagnosis was determined. For the six genes with high AUC (*RARB*, *KLLN*, *CHFR*, *TP73*, *GSTP1*, and *CASP8*), methylation scores (0 to 6) were defined based on their respective cut-off values to evaluate the detection of malignant groups by gene combinations. In addition, the sample size of the test set was determined with reference to the AUC of the training set. According to our statistical calculations, the required number of cases in the test set is 15. Therefore, we considered that using 16 samples in the test set is statistically valid. Fisher’s exact test was used to evaluate each gene and its prediction equation. *p* < 0.05 was considered significant. All statistical ROC analysis and Fisher’s exact test were performed using IBM SPSS Statistics version 26 (IBM Corporation). The sample size calculation was performed using G*Power version 3.1.9.7. (Universität Kiel, Kiel, Germany)

### 2.5. Cytology

Mucosal cells were collected by rubbing the tip of a brush against the lesion in the oral cavity. The samples collected were evaluated by Papanicolaou staining using the liquid-based cytology method.

## 3. Results

### 3.1. Clinicopathological Characteristics of Patients with OPMDs

Gargle fluid samples were collected from patients with clinically diagnosed OPMDs prior to biopsy. We prepared two independent sample sets, a training set and a test set, and verified the accuracy of the results obtained from the training set with those from the test set. The sample size for the test set was calculated based on the AUC obtained from the training set, and samples were collected from a target of 15 cases. The training set comprised 40 patients with clinical diagnoses of intractable stomatitis (n = 4), lichen planus (n = 15), and leukoplakia (n = 21). Pathological diagnosis showed malignant lesions in nine patients (22.5%), including intraepithelial carcinoma in the training set. The test set consisted of 16 patients with a clinical diagnosis of intractable stomatitis (n = 2), lichen planus (n = 5), and leukoplakia (n = 9), and pathological diagnosis showed malignant lesions in six patients (37.5%). There were no statistically significant differences in age, sex, smoking status, or alcohol consumption between the malignant and non-malignant groups (Table 1)

### 3.2. Diagnostic Performance of Each Gene on the Training Set

Using purified DNA from the gargle fluid, MS-MLPA was used to evaluate the methylation status of 25 tumor suppressor genes. The methylation level of each gene was calculated as methylation %, and ROC curves were generated with the detection of the malignant group from patients with OPMDs as the endpoint to evaluate the usefulness of each gene. Results of ROC analysis, nonparametric tests, Fisher test, and cutoff values for identification of malignant groups are shown (Table 2 and Appendix A, Figure 1, Appendix A). Of the 25 genes, 14 genes (*RARB*, *KLLN*, *CHFR*, *CADM1*, *TP73*, *GSTP1*, *BRCA1*, *ESR1*, *ATM*, *TIMP3*, *BRCA2*, *CASP8*, *MLH*, and *APC*) had an AUC in the range 0.6–0.7, and four genes (*RARB*, *KLLN*, *CHFR*, *CADM1*) had an AUC > 0.7. In addition, there were significant differences in Fisher’s exact test for the five genes (*RARB*, *KLLN*, *CHFR*, *CADM1*, and *TP73*), indicating their potential for detecting malignant groups. Similar to the 14 genes, the negative predictive value (NPV) was good, and the specificity was relatively good, but the sensitivity and positive predictive value (PPV) were poor.

### 3.3. Evaluation of a Unique Predictive Score for Early-Stage Cancer Identification

Because single gene evaluation did not have sufficient diagnostic performance to identify malignant groups, we performed a combination gene analysis to develop a more efficient diagnostic system to detect malignant groups. Using the cutoff value of each gene as an indicator of aberrant methylation, we evaluated the combination of a number of aberrantly methylated genes and found that the combination of six genes (*TP73*, *CASP8*, *RARB*, *KLLN*, *GSTP1*, and *CHFR*) was useful for identifying the malignant group. The diagnostic performance of the six gene combinations was AUC = 0.885, sensitivity = 77%, specificity = 87.1%, PPV = 63.6%, and NPV = 93% (Table 3, Figure 2). In addition, to evaluate the reproducibility of the score from the six genes, a unique prediction score based on a combination of six genes (*TP73*, *CASP8*, *RARB*, *KLLN*, *GSTP1*, and *CHFR*) for identifying malignant groups, we collected new samples from 16 patients clinically diagnosed with OPMDs for the test set. The diagnostic performance was AUC = 0.833, sensitivity = 66.7%, specificity = 80.0%, PPV = 66.7%, and NPV = 80.0% (Table 3, Figure 2), and we found the score from the six genes to be a reproducible method. Compared to single gene evaluation, the score from the six genes showed higher diagnostic performance for detecting malignant groups in clinically diagnosed OPMDs patients.

### 3.4. Comparing Cytology and the Score from the Six Genes

We compared the score from the six genes with those of the current commonly used cytology test. Of all the patients, 44 had a cytological diagnosis, of which only one patient could be identified as SCC (Table 4). Compared with the cytology tests, the score from the six genes in 34 patients (excluding 10 patients out of who exhibited an indefinite status for neoplasia) showed useful diagnostic accuracy in terms of sensitivity, PPV, and NPV (Table 5). For reference, the score from the six genes still showed high AUC and good test performance in the group of patients who underwent cytology (Figure 3).

## 4. Discussion

Our study showed that in patients with clinically diagnosed OPMDs, using gargle fluid to detect aberrant methylation might help to identify patients with early-stage oral cancer, including intraepithelial carcinoma. No previous studies have reported stepwise aberrant methylation detection using the gargle fluid of patients with OPMDs. Evaluation of the combination of six genes with aberrant methylation *(RARB*, *KLLN*, *CHFR*, *TP7*3, *GSTP1*, and *CASP8*) was particularly useful for identifying early-stage oral cancer in clinically diagnosed patients with OPMDs with high diagnostic performance. *RARB*, *KLLN*, *CHFR*, *TP73*, and *CASP8* have been reported in previous studies to be associated with methylation and OSCC in tissue samples and cell lines [32,33,34,35]. Although *GSTP1* has been reported to be associated with genetic polymorphisms and OSCC [36], we could not find any reports of an association between GSTP1 and methylation and OSCC; therefore, our results are novel. 

In our previous study, compared with the healthy group, we found abnormal methylation of *ECAD*, *TMEFF2*, *RARB,* and *MGMT* genes in the gargle fluid in the SCC group, which is useful for detecting patients with SCC [27]. To elucidate gene variation-associated multistage carcinogenesis, we examined patients clinically diagnosed with OPMDs for early cancer identification. Aberrant methylation of the *RARB* gene was observed in a previous study [27], suggesting that it may be associated with OSCC carcinogenesis. In future studies, it will be necessary to examine the relationship between OSCC progression and prognosis. In addition, methylation of *KLLN*, *CHFR*, *TP73*, *GSTP1*, and *CASP8* may be related to precancerous processes, such as epithelial hyperplasia and epithelial dysplasia. By examining the methylation of these genes, it is possible to evaluate the risk of oral cancer and to contribute to cancer prevention by preventing the induction of epigenetic abnormalities by environmental factors.

In four patients, the SCC could not be detected using the score from the six genes on the training and test sets. In one case, the tumor was in the root of the tongue, which may have made it difficult to collect tumor cells by gargling. In the remaining three cases, there were no obvious ulcers, and it was considered difficult to collect tumor cells due to the small area of lesion exposure. These are considered limitations of the gargle fluid test using aberrant DNA methylation.

Epigenetic factors such as HPV infection have been reported to alter DNA methylation, leading to malignant transformation of precancerous lesions [37]. However, HPV infection in the oral cavity is more uncommon than HPV infection in the oropharyngeal region [38]. In fact, the relationship between HPV infection causing malignant transformation of OPMDs and methylation of cancer-related genes in gargle fluid has never been investigated before. HPV infection is often observed in oropharyngeal cancer and is thus considered a therapeutic indicator. Therefore, the relationship between HPV infection and OPMDs, or HPV infection and DNA methylation in OPMDs should be examined in the future.

MS-MLPA can examine the methylation of multiple genes simultaneously using a probe mix. By comparing the digested sample with the non-digested sample, the methylation percentage can be calculated, and the quantification shows good performance.

In a previous study, the MultiNA microchip electrophoresis system was used for semi-quantitative analysis, which required processing with each primer and had a high running cost. The MS-MLPA method was able to analyze the methylation of multiple genes simultaneously, which was useful in terms of time and cost compared with the methods of previous studies.

Cytology testing to identify OSCC generally has high specificity, but low sensitivity [7]. The sensitivity of this test is dependent on proper sample collection and fixation procedures. Moreover, as cytology is diagnosed by a cytology diagnostician, the diagnosis varies according to the diagnostician. In fact, the sensitivity of cytology testing (brush biopsy) remains low at 18%. In particular, SCC patients were often determined to have LSIL and NILM. In contrast, the diagnostic performance of the score from six genes was better than that of cytology testing, with a sensitivity of 63.6%, and its specificity was similar to that of the cytology tests. This result may indicate that cytology testing requires precise scraping of local tumor cells, whereas gargling fluid is effective in collecting exfoliated cells of the oral mucosa, regardless of the location of the lesion. Therefore, we believe that the gargle test exhibits better diagnostic accuracy than a cytology test and is more convenient. In addition, the score can be determined by substituting the test value. Since neither a pathologist nor a diagnostician need to be present during the gargle test, it is considered to be a simpler screening tool than the cytology test, which is a pathological diagnostic test.

The gargle fluid test using DNA methylation can be used as a screening tool for patients with OPMD because it is safe and easy to collect specimens. In addition, because repeated samples can be easily collected, monitoring of the disease condition and progression may be possible. These results suggest that it can be used as a new, noninvasive screening tool. Although the sample size for this study was small, we confirmed the usefulness of the score from the six genes for both the training and test sets. In this study, only patients with early-stage oral cancer who were clinically diagnosed with OPMDs were included. Consequently, a limited number of patients diagnosed with early-stage OSCC were enrolled during the study period because the incidence of OSCC arising from OPMDs is approximately 10%. We acknowledge the small sample size as a limitation of our study. Nevertheless, the score from the six genes showed high AUC in the training set and good diagnostic accuracy. Therefore, we conclude that the score has the potential to help clinicians distinguish early-stage oral cancer from clinically diagnosed OPMDs. In addition, the sample size of the test set was determined with reference to the AUC of the training set. An AUC of 0.833 indicated that the diagnostic performance of the score was still acceptable in the test set; therefore, we considered the results from the training set to be validated. The aberrant methylation panel on the test set showed no significant difference in Fisher’s exact test but showed a tendency to be useful in detecting early-stage oral cancer due to the high AUC. We intend to examine more samples for practical application of the test in future studies.

## 5. Conclusions

In conclusion, we showed the possibility of distinguishing patients with early-stage cancer from patients clinically diagnosed with OPMDs by examining aberrant methylation of tumor suppressor genes in gargling fluid. Methylation combination scores have potential as a noninvasive and simple screening tool.

## Figures and Tables

**Figure 1 cancers-14-02646-f001:**
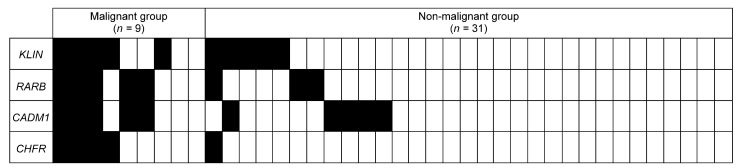
Methylation profiles are illustrated for the four genes with AUC over 0.7 in nine patients in the malignant group (pathologically diagnosed SCC and SCC intraepithelial patients) and in 31 patients in the non-malignant group (pathologically diagnosed epithelial dysplasia, epithelial hyperplasia, lichen planus, chronic stomatitis patients) in the training set. The map shows aberrant methylation (black boxes) or unmethylation (light boxes).

**Figure 2 cancers-14-02646-f002:**
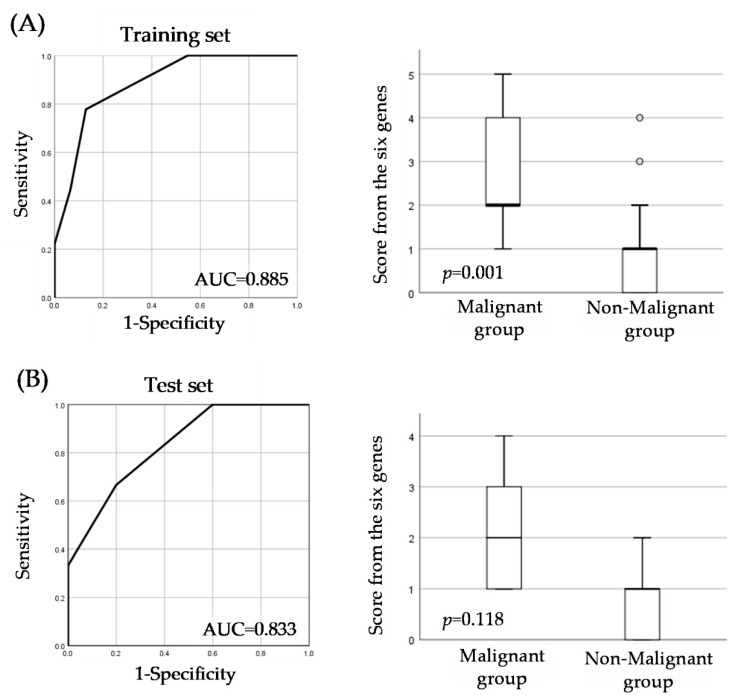
Diagnostic performance of the score from the six genes. (**A**) receiver operating characteristic (ROC) analysis and level of the score from the six genes on the training set. (**B**) ROC analysis and level of the score from the six genes on the test set.

**Figure 3 cancers-14-02646-f003:**
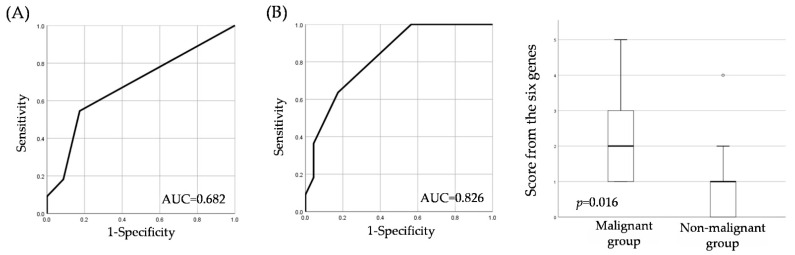
(**A**) ROC analysis of the cytology test group. (**B**) ROC analysis and level of the score from the six genes on patients with OPMDs who underwent cytology testing for detection of the malignant group.

**Table 1 cancers-14-02646-t001:** Clinicopathological characteristics of patients with OPMDs.

	Training Set			Test Set		
	Non-Malignant Group	Malignant Group		Non-Malignant Group	Malignant Group	
	*n* = 31	*n* = 9		*n* = 10	*n* = 6	
R	71 (30–86)	74 (37–81)		68.5 (45–83)	63 (24–87)	
**Sex, *n* (** **%)**						
Male	16 (51.6)	5 (55.5)	*p* = 1.000	4 (40.0)	3 (50.0)	*p* = 1.000
Female	15 (48.3)	4 (44.4)		6 (60.0)	3 (50.0)	
**Smoking, *n* (** **%)**						
Yes	11 (36.6)	4 (44.4)	*p* = 0.441	2 (20.0)	3 (50.0)	*p* = 0.299
No	20 (64.5)	5 (55.5)		8 (80.0)	3 (50.0)	
**Drinking, *n* (** **%)**						
Yes	10 (32.2)	4 (44.4)	*p* = 0.255	3 (30.0)	2 (33.3)	*p* = 1.000
No	21 (67.7)	5 (55.5)		7 (70.0)	4 (66.7)	
**Clinic diagnosis, *n* (** **%)**						
Intractable stomatitis	2 (6.4)	2 (22.2)		1 (10.0)	1 (16.7)	
Lichen planus	13 (41.9)	2 (22.2)		4 (40.0)	1 (16.7)	
Leucoplakia	16 (51.6)	5 (55.5)		5 (50.5)	4 (66.7)	
**Pathological diagnosis, *n* (** **%)**						
Chronic stomatitis	12 (38.7)			2 (20.0)		
Lichen planus	2 (6.4)			2 (20.0)		
Epithelial hyperplasia	6 (19.3)					
Epithelial dysplasia	11 (35.4)			5 (50.0)		
Squamous cell papilloma				1 (10.0)		
SCC intraepithelial		4 (44.4)			1 (16.7)	
SCC		5 (55.5)			5 (83.3)	

Abbreviations: OPMDsS, oral potentially malignant disorders; SCC, squamous cell carcinoma. Nominal variables are indicated by *n* values.

**Table 2 cancers-14-02646-t002:** Diagnostic performance of each gene on training set.

Gene Name	Sensitivity	Specificity	PPV	NPV	Fischer‘s Exact Test	AUC
	%	%	%	%	*p*-Value	
*RARB*	55.6	90.3	62.5	87.5	0.008	0.731
*KLLN*	55.6	83.9	50.0	86.7	0.029	0.708
*CHFR*	44.4	96.8	80.0	85.7	0.006	0.708
*CADM1*	55.6	83.9	50.0	86.7	0.029	0.703
*TP73*	55.6	83.9	50.0	86.7	0.029	0.690
*GSTP1*	22.2	96.8	66.7	81.1	0.121	0.685
*BRCA1*	33.3	96.8	75.0	83.3	0.030	0.676
*ESR1*	77.8	61.3	36.8	90.5	0.060	0.672
*ATM*	44.4	83.9	44.4	83.9	0.168	0.658
*TIMP3*	44.4	80.6	40.0	83.3	0.190	0.652
*BRCA2*	44.4	83.9	44.4	83.9	0.168	0.629
*CASP8*	44.4	71.0	30.8	81.5	0.437	0.627
*MLH*	55.6	64.5	31.3	83.3	0.441	0.609
*APC*	33.3	87.1	42.9	81.8	0.316	0.606

**Table 3 cancers-14-02646-t003:** Diagnostic performance of the score from the six genes on the training and test sets.

Sample Set	Cutoff *	Malignant Group	Non-Malignant Group	Sensitivity	Specificity	PPV	NPV	Fischer’s Exact Test	AUC
				%	%	%	%	*p*-Value	
Training set (*n*= 40)	≥2 genes	7	4	77.8	87.1	63.6	93.1	0.001	0.885
	<2 genes	2	27						
Test set (*n* = 16)	≥2 genes	4	2	66.7	80.0	66.7	80.0	0.118	0.833
	<2 genes	2	8						

* The cutoff value represents the number of positive genes among six genes (*TP73*, *CASP8*, *RARB*, *KLLN*, *GSTP1*, and *CHFR*).

**Table 4 cancers-14-02646-t004:** Characteristics of patients who underwent cytology testing. HSIL, high grade squamous intraepithelial lesion. LSIL, low grade squamous intraepithelial lesion. NLIM, negative for intraepithelial lesion or malignancy.

		Biopsy			
		SCC	Dysplasia	Others	
Cytology					
	SCC	1			*p* = 0.580
	HSIL	1	2		
	LSIL	4		2	
	NILM	5	8	11	
Score from the six genes					
	≥2 genes	7	2	2	*p* = 0.016
	<2 genes	4	8	11	

**Table 5 cancers-14-02646-t005:** Diagnostic performance of cytology tests and the score from the six genes. In cytology, SCC and HSIL were classified as positive groups, whereas LSIL and NILM were classified as negative groups, because when cytology diagnoses SCC and HSIL, we consider that the lesion is a target for biopsy or resection.

Diagnostic Method	Cutoff	Malignant Group	Non-Malignant Group	Sensitivity	Specificity	PPV	NPV	Fischer’s Exact Test	AUC
				%	%	%	%	*p*-Value	
Cytology	SCC and HSIL	2	2	18.2	91.3	50.0	70.0	0.580	0.682
	LSIL and NILM	9	21						
Score from the six genes	≥2 genes	7	4	63.6	82.6	63.6	82.6	0.016	0.826
	<2 genes	4	19						

## Data Availability

The data presented in this study are available upon request from the corresponding author.

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
