# Peer review of "Detecting Early-Stage Oral Cancer from Clinically Diagnosed Oral Potentially Malignant Disorders by DNA Methylation Profile"

_cancers, 2022, doi:10.3390/cancers14112646_

Round 1
Reviewer 1 Report
In this study, the authors claim to have identified 6 genes with aberrant methylation that could distinguish between early-stage oral cancer from oral potentially malignant disorders. In addition, they also claim that it more accurately detected the presence of early stage oral cancer compared to cytology. This study was interesting to read but the small sample sizes used, and the lack of secondary/validation cohort leads my to recommend major revisions prior to acceptance of this manuscript.
- The most obvious issue is the small sample size used, and that only a training and testing cohort was utilized. Before making claims that this is superior to other methods, I think it is crucial that this be tested on a validation/secondary cohort, to see if it holds. Also, they mention that for 4 patients this signature didn't work. This impacts the small sample size
2. The other issue is of HPV status. How would this signature perform if there is a mixed sample of early stage oral cancers that are caused by HPV and others caused by the different aetiologies such as excessive smoking and drinking? Would it still hold? This would be a valid question since we know HPV can cause oral cancers, and that the HPV oncoproteins can perturb the methylation status of many different genes that can lead to a slightly different methylation profile between HPV+ and HPV- carcinomas.
3. Line 141: Ligation is repeated.
4. Figure 2 Legend; (A) is repeated twice.
Author Response
We thank you for your thoughtful suggestions and insights. We have prepared a response. Please see the attachment.

Reviewer 2 Report
The manuscript describes a study on methylation profile in early-stage oral cancer and OPMDs using non- invasive oral gargle fluid It is well written and presented and extends previous work focused on oral cancer.
However, a major issue is that the numbers of patients in the non-malignant group (n=31) and the malignant group (n=9) were quite unbalanced. How did the authors account for this when developing their diagnostic model?
Please add more discussion on the implementation of the methylation test as a screening tool. The authors state that ‘The gargle fluid test using DNA methylation can be used as a screening tool for patients with OPMD because it is safe and easy to collect specimens and can be used in non-specialized facilities.’ Certainly gargle fluid is easy to collect in a non-specialised facility but so is brush biopsy for cytology. Please include further discussion on the MS-MLPA method and whether specialised facilities are required.
Minor comments
- Check spelling in 2.4. Statical analysis
- Check Table 1 Abbreviations: OPMDS
- Figure 2A and B and figure 3B – indicate whether score is significantly different for malignant vs. non-malignant group.
- Although GSTP1 has been reported to be associated with genetic polymorphisms and OSCC [30], we could not find any reports of an association between GSTP1 and methylation and OSCC; therefore, our results are rare. Change ‘rare’ to ‘novel’.
Author Response

(The authors gave the same response as above.)

Reviewer 3 Report
For authors:
This manuscript provided a comprehensive analysis of current knowledge in this field. It shows rich content and I found it to be well-written and accessible. However, the major concern of this manuscript is with the introductive section and the results section.
Introduction:
- Reference number 1 (Siegel) should be updated to the most current one (from 2022).
- From line 50 to 64 there is no reference!!! References are needed in those sentences.
- I do not find the following reasoning logical: “OPMDs are difficult to diagnose based on clinical findings alone because of the wide variety of clinical conditions, and invasive biopsy is necessary to obtain a definitive diagnosis, which can only be performed in specialized facilities. Cytology is also widely used as a screening test; however, similar to biopsy, it requires diagnosis at a specialized facility. In addition, the diagnostic accuracy of cytology has high specificity, but low sensitivity [2]. Owing to these problems, there is a need for a simple and highly accurate testing tool that does not require specialization. “
Why do you say that it does not require specialization? It is logical that the diagnosis and follow-up of these lesions should be done in specialized centers and with qualified personnel. It is necessary to know how to take the sample, in which patients, how to interpret the results... This subject is not touched upon again in the article in the discussion part so I understand that it does not have much relevance.
- We recommend adding this article to the introduction: State of methylation and epigenome studies in some types of head and neck cancer (Bonar-Alvarez P, Perez-Sayans M, Padín-Iruegas E, Suarez-Penaranda JM, Oliveira- Alves M, Blanco Carrion A and García-Garcia A)
- OPMDs are not clearly defined. We recommended this article: Oral potentially malignant disorders: A consensus report from an international seminar on nomenclature and classification, convened by the WHO Collaborating Centre for Oral Cancer (Saman Warnakulasuriya , Omar Kujan , José M Aguirre-Urizar , José V Bagan, Miguel Ángel González-Moles , Alexander R Kerr , Giovanni Lodi , Fernanda Weber Mello, Luis Monteiro, Graham R Ogden, Philip Sloan, Newell W Johnson)
Methods:
There is a specific and detailed explanation for the majority of methods used in this study. However, the methods used to perform cytology are not described.
Results:
- What is most striking is that Intractable stomatitis is NOT an OPMD. If you can justify that it is with current evidence it is correct, otherwise, those cases would not make sense to include them as OPMDs.
- Lines 173 and 174 indicate that there are no significant differences in the variables in Table 1, however, p values are missing in the table.
- The data in rows 219-220 are repeated in Table 3 and Figure 2.
- Figure 2. The legend is incorrect, two (A)
- Line 239: ¿IFN? ¿What is IFN?
- Table 4, p value is missing
- In Table 5, the 2nd part of the legend refers to what? Positive group of what?

Author Response

(The authors gave the same response as above.)

Round 2
Reviewer 2 Report
Thanks to the authors for addressing my concerns
Reviewer 3 Report
The authors have improved significantly the manuscript according to the suggestions. I recommend to publish it in the present form.